

# Hygiene of housing conditions and proinflammatory signals alter gene expressions in porcine adipose tissues and blood cells

Audrey Quéméner, Marie-Hélène Perruchot, Frédéric Dessauge, Annie Vincent, Elodie Merlot, Nathalie Le Floch and Isabelle Louveau

PEGASE, INRAE, Institut Agro, Saint Gilles, France

Corresponding author
Isabelle Louveau,
isabelle.louveau@inrae.fr

## ABSTRACT

Adipose tissue is an organ with metabolic, endocrine and immune functions. In this tissue, the expressions of genes associated with several metabolic pathways, including lipid metabolism, have been shown to be affected by genetic selection for feed efficiency, an important trait to consider in livestock. We hypothesized that the stimulation of immune system caused by poor hygiene conditions of housing impacts the molecular and cellular features of adipose tissue and that the impact may differ between pigs that diverge in feed efficiency. At the age of 12 weeks, Large White pigs from two genetic lines divergent for residual feed intake (RFI) were housed in two contrasting hygiene conditions (good vs poor). After six weeks of exposure, pigs were slaughtered ($n = 36$). Samples of blood, subcutaneous (SCAT) and perirenal (PRAT) adipose tissues were collected for cell response and gene expression investigations. The decrease in the relative weight of PRAT was associated with a decline in mRNA levels of *FASN*, *ME*, *LCN2* and *TLR4* ($P < 0.05$) in pigs housed in poor conditions compared with pigs housed in good conditions for both RFI lines. In SCAT, the expressions of only two key genes (*PPARG* and *TLR4*) were significantly affected by the hygiene of housing conditions. Besides, the mRNA levels of both *LCN2* and *GPX3* were influenced by the RFI line ($P < 0.05$). Because we suspected an effect of poor hygiene at the cellular levels, we investigated the differentiation of stromal vascular cells isolated from SCAT *in vitro* in the absence or presence of a pro-inflammatory cytokine, Tumor Necrosis Factor-$\alpha$ (TNF-$\alpha$). The ability of these cells to differentiate in the absence or presence of TNF-$\alpha$ did not differ among the four groups of animals ($P > 0.05$). We also investigated the expressions of genes involved in the immune response and lipid metabolism in whole blood cells cultured in the absence and presence of LPS. The hygiene conditions had no effect but, the relative expression of the *GPX3* gene was higher ($P < 0.001$) in high RFI than in low RFI pigs while the expressions of *IL-10* ($P = 0.027$), *TGFβ1* ($P = 0.023$) and *ADIPOR2* ($P = 0.05$) genes were lower in high RFI than in low RFI pigs. Overall, the current study indicates that the hygiene of housing had similar effects on both RFI lines on the expression of genes in adipose tissues and on the features of SCAT adipose cells and whole blood cells in response to TNF-$\alpha$ and LPS. It further demonstrates that the number of genes with expression impacted by housing conditions was higher in PRAT than in SCAT. It suggests a depot-specific response of adipose tissue to the current challenge.

## INTRODUCTION

Farm animals like pigs are exposed to multiple environmental stimuli during their growth (*Colditz & Hine, 2016*), some of them have been clearly identified to be risk factors of poor health. For example, poor hygiene of housing results in degraded health, reduced growth performance, and immune system activation, characterized by a low-grade inflammation (*Chatelet et al., 2018*). This inflammatory response relies on the production of proinflammatory cytokines such as interleukin-6 (IL-6) or tumor necrosis factor α (TNF-α), which contribute to the redirection of nutrients to the immune system at the expense of muscle and adipose tissues (*Gabler & Spurlock, 2008*; *Ajuwon, 2014*).

Initially, white adipose tissue was considered as a preferential site of energy storage in the form of triacylglycerols (*Ameer et al., 2014*). Now, it is also recognized as an endocrine organ that produces a large number of secretory products termed adipokines or adipocytokines, either from adipocytes or from cells of the stromal vascular fraction, which contains preadipocytes, lymphocytes, macrophages, fibroblasts and vascular cells (*Louveau et al., 2016*). These adipokines, such as leptin, adiponectin, insulin-like growth factor-I (IGF-I) but also IL-6, IL-10 or TNF-α, communicate with other organs including brain, liver, skeletal muscle, the immune system, and adipose tissue itself (*Fantuzzi, 2005*; *Ramsay & Caperna, 2009*). Their expression can depend on the anatomical location (subcutaneous or visceral) of adipose tissues (*Tchkonia et al., 2013*). In addition, an increase in body fat mass may trigger the development of an inflammatory state (*Stolarczyk, 2017*). Therefore, the biological processes underlying the contribution of adipose tissue to an inflammatory response and hence the ability of pigs to sustain sanitary challenges deserves further studies (*Patience, Rossoni-Serão & Gutiérrez, 2015*).

Selection of pigs for residual feed intake (RFI) has been used to improve feed efficiency. Briefly, for a similar production level, high RFI (HRFI) pigs eat more than predicted based on average requirements for growth and maintenance and therefore are less efficient than low RFI (LRFI) pigs (*Gilbert et al., 2007*; *Gilbert et al., 2017*). Genetic selection for RFI results in changes in the partition of nutrients between maintenance and growth (*Labussière et al., 2015*; *Merlot, Gilbert & Le Floc'h, 2016*). With respect to adipose tissue, this selection induces changes in the expression of genes involved in several pathways including pathways related to lipid metabolism and defense response (*Gondret et al., 2017*; *Horodyska et al., 2019b*). These differences in nutrient partitioning may alter pig ability to allocate nutrients for stress and immune responses when facing environmental challenges (*Rauw et al., 1998*). For example, it was reported that LRFI growing pigs coped better with an immune challenge caused by poor hygiene of the housing than HRFI pigs (*Chatelet et al., 2018*). We hypothesized that this difference between the two RFI lines may involve changes in the molecular and/or cellular features of adipose tissues and blood immune cells. Therefore, the present study was carried out to evaluate the effects of poor
hygiene of housing conditions on the expression of genes in adipose tissue of LRFI and HRFI pigs. *In vitro*, we reproduced inflammatory conditions using an exposure to TNF-α or lipopolysaccharide (LPS), the main structural component of the outer membrane of most Gram-negative bacteria, to determine the impact of inflammatory stimuli and genetic selection on adipocyte differentiation and on the expression of genes involved in immune-adipose communication in blood immune cells.

## MATERIALS & METHODS

### Ethical approval/declaration

The experiment was performed in the INRAE UE3P experimental facility (Saint-Gilles, France; https://doi.org/10.15454/1.5573932732039927E12) in compliance with the ethical standards of the European Community (Directive 2010/63/EU) and was approved by the regional ethical committee (Comité Rennais d'Ethique en matière d'expérimentation animale or CREEA Rennes). The experiment approval number is APAFIS#494–2015082717314985.

### Animals and sample collection

For this experiment, a subset of growing Large White pigs (males and females) was selected from a larger study previously described by *Chatelet et al. (2018)*. Pigs originated from the 8th generation of two lines divergently selected for residual feed intake (RFI), a measure of feed efficiency, were used (*Gilbert et al., 2007*). They were weaned at four weeks of age. At 12 weeks of age, same-sex pairs of pigs were selected within a litter from the high RFI (HRFI, $n = 16$) and the low RFI (LRFI, $n = 20$) lines. Within a pair, animals were then randomly assigned to a room with good hygiene conditions ($n = 8$ and 10, respectively) and the other half to a room with poor hygiene conditions ($n = 8$ and 10, respectively). Good housing conditions included room cleaning, disinfection, and adoption of strict biosecurity precautions. In contrast, poor hygiene conditions consisted of no cleaning nor sanitation of the room after the previous occupation by non-experimental pigs and during the experiment as fully described previously by *Chatelet et al. (2018)*. This model is known to induce the stimulation of body defenses (inflammation, immune response, oxidative status) that is not caused by a specific and single biotic (virus, bacteria, fungi...) or abiotic (gas, dust...) agents. All pigs were housed in individual pens (85 × 265 cm) on concrete floor and have free access to a standard diet formulated to meet or exceed the nutritional requirement of growing pigs (9.47 MJ of net energy/kg, starch 44.2%; fat 3.1%; crude protein 15.3%), and to water. After six weeks in these housing conditions and after an overnight fast, animals were euthanized by electrical stunning and exsanguination. Immediately after slaughter, blood samples were collected from the jugular vein on heparin. Samples of subcutaneous dorsal adipose tissue (SCAT) and perirenal adipose tissue (PRAT) were collected, cut into small pieces, frozen in liquid nitrogen, and stored at −75 °C until total RNA extraction. For cell isolation from SCAT, portions of tissues were placed in warm Krebs-Ringer-bicarbonate-HEPES buffer and processed within 30 min (*Perruchot et al., 2013*).

## Culture of whole blood cells

Whole blood cells were incubated in resting conditions (medium alone) and after activation with LPS (O55:B5; Sigma-Aldrich, St. Quentin Fallavier, France). This molecule is present on the wall of Gram-negative bacteria and induces the production of inflammatory mediators by immune cells via the activation of Toll-like receptor 4 (TLR4). Whole heparinized blood samples were diluted 1:5 in complete cell culture medium (RPMI supplemented with 10% fetal bovine serum (FBS), 1% L-Glutamine, and 1% penicillin/streptomycin), and distributed in 24-well culture plates (0.4 mL/well) in triplicates. Complete medium (0.6 mL) alone or supplemented with LPS (Sigma, 10 µg/mL in the well) was then added. After 20 h of incubation in 95% air-5% $CO_2$ at 37 °C, cells were collected, triplicates were pooled, spun down, suspended in 0.5 mL of lysis buffer DL (Macherey-Nagel, Hoerdt, France) and stored at −75 °C.

## Adipose cell isolation

Cells were isolated from fresh subcutaneous dorsal adipose tissue by collagenase digestion as previously described (*De clercq et al., 1997*; *Perruchot et al., 2013*). Briefly, fresh adipose tissue was cut into small pieces and dissociated by enzymatic digestion with collagenase II and XI (Sigma, St. Quentin Fallavier, France) under shaking in a dry bath for 45 min at 37 °C. Then, a centrifugation at 400 g for 10 min was performed to separate floating adipocytes from the pellet of stromal vascular fraction (SVF) cells. After resuspension, SVF cells were successively filtered through 200-µm and 25-µm nylon membranes (Dutscher, Brumath, Alsace, France). After isolation, SVF cells were placed in FBS/10% dimethyl sulfoxide (DMSO; Sigma) and frozen at −150 °C.

## Culture of stromal vascular fraction cells and differentiation induction

First, individual vials of $1 \times 10^6$ cryopreserved SVF cells were thawed in a 37 °C water bath (2–3 min). Then, cells were grown at 37 °C under 95% air-5% $CO_2$ and were first placed in Dulbecco's modified Eagle's medium (DMEM 1X, GlutaMAX; with 4,5 g/L glucose; Gibco, Invitrogen, USA) supplemented with 20% FBS and 50 U/mL Penicillin-Streptomycin (Gibco, Invitrogen) until confluence in T25 cell culture flask (Nunc EasYFlask 25 cm$^2$; Thermo Fisher Scientific, France). Then, at day 0 (D0), cells were seeded in six-well plates (NUNCLON$^{TM}$ Delta Surface; Thermo Fisher Scientific), at a density of $2.8 \times 10^5$ cells/well (9.6 cm$^2$/well), coated with growth factor reduced Matrigel (1/50 vol/vol; Corning, Boulogne-Billancourt, France) as described previously (*Perruchot et al., 2013*). From D4 to D6 of culture, the medium was changed and replaced by DMEM, GlutaMAX, high glucose (Gibco, Invitrogen) supplemented with 2.5% swine serum (Gibco, Invitrogen), 50 U/mL Penicillin-Streptomycin (Gibco, Invitrogen), 2.6 nM porcine insulin (Sigma, Lyon, France) and 100 nM Cortisol (Sigma). On D6, cells were placed in a medium allowing adipocyte differentiation. This culture medium consisted of DMEM, GlutaMAX, high glucose supplemented with 50 U/mL Penicillin-Streptomycin, 100 nM 3-Isobutyl-1-methylanxthine (IBMX; Sigma), 2.6 nM insulin, 10 µM dexamethasone (Sigma), 0.2 nM T3 (Sigma), 10 µg/mL transferrin (Sigma), 100 µM ascorbic acid (Sigma) and 100 nM Cortisol (Sigma). The differentiation culture medium was renewed every two to three days. From

D4 to D22 of culture, cells seeded to study differentiation were treated without (control condition) or with 1 ng/mL or 10 ng/mL of recombinant porcine TNF-α (Invitrogen).

In order to assess the percentage of cell differentiation, pictures were taken on D14, D18 and/or D22 of culture using a phase contrast microscope at ×40 magnification (AxioVert A1; Zeiss, Oberkochen, Germany). Lipid accumulation in cells was estimated from two wells per condition tested for each animal and three random fields per well with ImageJ software (ImageJ 1.50i; National Institutes of Health, Bethesda, MD, USA). Results were expressed as surface ratio of lipid droplets present in the cells divided by the total area of cells in the same field.

## Total RNA extraction from whole blood cells and adipose tissues

For whole blood cells, total RNAs were extracted using a commercial kit (NucleoSpin RNA blood kit; Macherey-Nagel) according to the manufacturer's instructions. After a DNase treatment (DNA-free kit; Applied Biosystems, Foster City, CA, USA) in the presence of a RNase inhibitor (Thermo Fisher Scientific), the quality and amount of extracted RNA were estimated using a microvolume DS-11 spectrophotometer (DeNovix, Clinisciences, France). After a concentration step performed with a speed-vac concentrator (Thermo Fisher Scientific), only samples reaching the minimal required RNA concentration of 111 ng/μL were used. For both adipose tissues, total RNAs were extracted using RNeasy Lipid Tissue Mini Kit (Qiagen, France) according to the manufacturer's instructions, and were quantified using the DS-11 spectrophotometer. The integrity of extracted total RNA from whole blood cells and adipose tissues was assessed using the RNA 6000 Nano kit (Agilent Technologies, Paris, France) with the Agilent 2100 Bioanalyzer (Agilent Technologies) and samples meeting quality criteria were kept for further analyses. Ratios of A260/280 and A260/230 were greater than 1.6. RNA integrity numbers were between 7.0 and 10.0.

## Quantitative real-time PCR (qPCR)

For the determination of the expression levels of genes in blood cells and in adipose tissues, first-strand cDNA synthesis was performed with 1 μg of total RNA, by using High Capacity RNA to cDNA kit according to the manufacturer's instructions (Applied Biosystems, Foster City, CA, USA). Primers (Table 1) were designed from porcine sequences available in Ensembl or NCBI databases using Primer Express® v3.0 software (Applied Biosystems).

For blood cell samples, amplification reactions and disassociation curves were carried out on a Step One Plus™ real-time PCR system (Applied Biosystems). Among the seven tested house-keeping genes (*B2M*, *GAPDH*, *HRPT1*, *PPIA*, *TBP1*, *YWHAZ*, *18S*), *B2M*, *PPIA* and *TBP1* were used to calculate the normalization factor. They were identified as the most stable house-keeping genes by the RefFinder algorithm (https://github.com/SEAL-UCLA/Ref-Finder) and their expression cycle threshold was not influenced by the effects of line and housing conditions.

Expression levels of genes in adipose tissues were determined using the SmartChip Real-Time PCR system with a 5184-well chip (TaKaRa, Saint-Germain-en-Laye, France) available at the EcogenO Genomic Platform (https://ecogeno.univ-rennes1.fr/; OSUR, Rennes, France). Amplification reactions were carried out using LightCycler 480 SYBR

**Table 1  Primer sequences used for analysis of gene expression by qPCR.**

| Gene symbol | Gene name | Accession number[a] | Primer and probe sequences |
|---|---|---|---|
| ACOX1 | Acyl-CoA oxidase 1 | XM_021066020.1 | F: TAGCCCTACTGTGACTTCCATCAA |
| | | | R: GCCAGTACTATCGCGTGATTTG |
| ADIPOQ | Adiponectin | NM_214370.1 | F: GCTGTACTACTTCTCCTTCCACATCA |
| | | | R: CTGGTACTGGTCGTAGGTGAAGAGT |
| ADIPOR1 | Adiponectin receptor 1 | NM_001007193.1 | F: GCCATGGAGAAGATGGAGGA |
| | | | R: AGCACGTCGTACGGGATGA |
| ADIPOR2 | Adiponectin receptor 2 | XM_021091197.1 | F: TGTTCGCCACCCCTCAGTAT |
| | | | R: AATGATTCCACTCAGGCCCA |
| CEBPA | CCAAT/enhancer binding protein alpha | XM_003127015.4 | F: GTGGACAAGAACAGCAACGA |
| | | | R: CTCCAGCACCTTCTGTTGAG |
| COX1 | Mitochondrial cytochrome c oxidase subunit 1 | XM_021064446.1 | F: GAATAGTGGGCACTGCCTTGA |
| | | | R: GGGTTCCGGGCTGACCTA |
| COX3 | Mitochondrial cytochrome c oxidase subunit 3 | AF034253.1 | F: CGTCCCATCCCTCGGTTTA |
| | | | R: GCCAGGTCGTGTGGATATTAGAGT |
| CS | Citrate synthase | XM_021091147.1 | F: CCTTTCAGACCCCTACTTGTCCT |
| | | | R: CACATCTTTGCCGACTTCCTTC |
| DLK1 | Delta like non-canonical Notch ligand 1 | NM_001126101.2 | F: CCCATGGAGCTGAATGCCT |
| | | | R: TTGCAAATGCACTGCCAGGG |
| ME | Malic enzyme 1 | XM_001924333.5 | F: TGGTGACTGATGGAGAACGTATTC |
| | | | R: CAGGATGACAGGCAGACATTCTT |
| FABP4 | Fatty acid binding protein 4 | NM_001002817.1 | F: GGAAAGTCAAGAGCACCATAACCT |
| | | | R: ATTCCACCACCAACTTATCATCTACTATTT |
| FASN | Fatty acid synthase | NM_001099930.1 | F: AGCCTAACTCCTCGCTGCAAT |
| | | | R: TCCTTGGAACCGTCTGTGTTC |
| GLUT4 | Glucose transporter 4 | NM_001128433.1 | F: GGCAGCCCCTCATCATTG |
| | | | R: TCGAAGATGCTGGTTGAATAGTAGAA |
| GPX3 | Glutathione peroxidase 3 | NM_001115155.1 | F: GCTTCCCCTGCAACCAATT |
| | | | R: GGACATACCTGAGAGTGGACAGAA |
| LIPE | Lipase E hormone sensitive | NM_214315.3 | F: CAACTTGGTGCCCACAGAAGA |
| | | | R: GTCATGCAGTGTCAGGTACTTGAGA |
| IGF1 | Insulin-like growth factor 1 | NM_214256.1 | F: GCTGGACCTGAGACCCTCTGT |
| | | | R: TACCCTGTGGGCTTGTTGAAAT |
| IGF2 | Insulin-like growth factor 2 | NM_213883.2 | F: AGGGCATCCAAACCACAAAC |
| | | | R: GGGTTCAATTTTTGGTATGTAACTTG |
| IL-1 β | Interleukin 1beta | NM_214055.1 | F: GCCAGTCTTCATTGTTCAGGTTT |
| | | | R: TTGTCACCGTAGTTAGCCATCACT |
| IL-6 | Interleukin 6 | NM_001252429.1 | F: CTGGCAGAAAACAACCTGAACC |
| | | | R: TGATTCTCATCAAGCAGGTCTCC |
| IL-10 | Interleukin 10 | NM_214041.1 | F: TGAGAACAGCTGCATCCACTTC |
| | | | R: TCTGGTCCTTCGTTTGAAAGAAA |

| Gene symbol | Gene name | Accession number[a] | Primer and probe sequences |
|---|---|---|---|
| IL-15 | Interleukin 15 | XM_021100480.1 | F: TGCATCCAGTGCTACTTGTGTT |
|  |  |  | R: TTAGGAAGACCTGCACTGATACAG |
| LCN2 | Lipocalin 2 | XM_021088538.1 | F: TCGCAATCGACCAGTGCAT |
|  |  |  | R: TGGGCAAAGGCTGAAGACAT |
| LEPTIN | Leptin | XM_021078503.1 | F: GTTGAAGCCGTGCCCATCT |
|  |  |  | R: CTGATCCTGGTGACAATCGTCTT |
| LEPR | Leptin Receptor | NM_001024587.1 | F: CAGTCGCTCAGTGCTTATCCT |
|  |  |  | R: GGAAGGGATTCTGAGCCATT |
| PPARG | Peroxisome proliferator activated receptor gamma | XM_005669788.3 | F: ATTCCCGAGAGCTGATCCAA |
|  |  |  | R: TGGAACCCCGAGGCTTTAT |
| SOD2 | Superoxide dismutase 2 | NM_214127.2 | F: GCGCTGAAAAAGGGTGATGT |
|  |  |  | R: ACCGTTAGGGCTCAGATTTGTC |
| SREBP1 | Sterol regulatory element binding transcription factor 1 | XM_021066226.1 | F: CGGACGGCTCACAATGC |
|  |  |  | R: GCAAGACGGCGGATTTATTC |
| TGF β1 | Transforming growth factor beta 1 | XM_021093503.1 | F: AGCGGCAACCAAATCTATGATAA |
|  |  |  | R: CGACGTGTTGAACAGCATATATAAGC |
| TLR2 | Toll like receptor 2 | XM_005653577.3 | F: CCCAGCACAGTGATGAAAAAATT |
|  |  |  | R: TCCGTAAAACTTGCGTCAGTGA |
| TLR4 | Toll like receptor 4 | NM_001113039.2 | F: AACATCCCCACATCAGTCAAGAT |
|  |  |  | R: CCCTGATATGCATCATCGTCAA |
| TNF-α | Tumor necrosis factor alpha | NM_214022.1 | F: GGTTATCGGCCCCCAGAA |
|  |  |  | R: TGGGCGACGGGCTTATC |
| UCP3 | Uncoupling protein 3 | NM_214049.1 | F: AAGTACAGCGGGACGATGGA |
|  |  |  | R: TGTTGGGCAGAATTCCTTTCC |
| B2M | Beta-2-microglobulin | NM_213978 | F: AAACGGAAAGCCAAATTACC |
|  |  |  | R : ATCCACAGCGTTAGGAGTGA |
| PPIA | Peptidylprolyl isomerase A | XM_021078519.1 | F: AGCACTGGGGAGAAAGGATT |
|  |  |  | R: AAAACTGGGAACCGTTTGTG |
| TBP1 | TATA box binding protein | DN110073 | F: AACAGTTCAGTAGTTATGAGCCAGA |
|  |  |  | R : AGATGTTCTCAAACGCTTCG |
| YWHAZ | Tyrosine 3-monooxygenase/tryptophan 5-monooxygenase activation protein zeta | XM_005662949.2 | F: ATGCAACCAACACATCCTATC |
|  |  |  | R: GCATTATTAGCGTGCTGTCTT |

**Notes.**

[a]Accession number in the National Center for Biotechnology Information database (http://www.ncbi.nlm.nih.gov/gene) or Ensembl project database (http://www.ensembl.org/Sus_scrofa/Info/Index) for pig sequences.

[b]F, forward primer

[c]R, reverse primer

[d]Genes used as reference for qPCR normalization.

Green 1 Master (Roche Diagnostics, Bagnolet, Ile-de-France, France) with a final cDNA concentration of 1 ng/μL and a primer concentration of 500 nM dispensed using the SmartChip Multisample Nanodispenser. Amplification conditions were as follows: 5 min at 95 °C followed by 45 cycles of 30 s at 95 °C, 30 s at 60 °C and 30 s at 72 °C, followed by 15 s at 95 °C and 1 min at 60 °C. Specificity of the amplification products was checked by dissociation curve analysis. *YWHAZ* and *PPIA* were identified as the most stable housekeeping genes and were used for normalization. For all the examined genes, the normalized expression level N was calculated according to the following formula: $N = E^{-\Delta Cq(sample-calibrator)}/NF$ where E is calculated from the slope of calibration curve, Cq is the quantification cycle, calibrator is a pool of all samples and NF is the normalized factor calculated using the geNorm algorithm. For all studied genes, E was between 1.68 and 2.10.

## Statistical analyzes

Data were subjected to analysis of variance (ANOVA) considering the individual pig as the experimental unit. For data related to adipose tissues, statistical analyzes were performed using the GraphPad Prism software (version 8.4.3 (686); GraphPad, San Diego, CA, USA). For data related to gene expressions, hygiene conditions, RFI line and their interactions were considered as fixed effects. For data related to the response of adipose cells to TNF-α, hygiene conditions, RFI line, TNF-α dose and their interactions were considered as fixed effects. For whole blood cells, hygiene conditions, RFI line, presence of LPS and their interactions were considered as fixed effects, the pig was used as a random factor, and ANOVA was performed using R software (version 3.5.1; *R Core Team, 2018*). Results are presented as mean ± standard error of mean (SEM). Differences were considered statistically significant if $P \leq 0.05$ and were discussed as a trend if $0.05 < P < 0.1$.

## RESULTS

### Growth performance and body composition

Growth performance and body composition of selected pigs have been previously described in *Sierzant et al. (2019)*. Briefly, average daily gain was significantly lower in HRFI pigs housed in poor hygiene conditions than in HRFI pigs housed in good hygiene conditions with no significant difference between LRFI pigs. The relative weight of PRAT was significantly lower in poor hygiene pigs than in good hygiene pigs for both RFI lines. The proportion of SCAT in both RFI lines was not influenced by hygiene of housing conditions.

### Expression of genes in adipose tissues

The relative expressions of genes related to adipocyte differentiation, metabolism, secretory functions, oxidative stress and pathogen recognition, were determined in perirenal (PRAT) and subcutaneous (SCAT) adipose tissues. Data related to genes with relative expressions significantly affected by housing conditions and/or RFI selection in PRAT and/or in SCAT are shown in Tables 2 and 3. Other data are presented in Table S1 and S2. First, there was no significant interaction between housing conditions and RFI lines (Tables 2 and 3) with the exception of the *COX1* gene in SCAT.

**Table 2  Relative expression of genes in perirenal adipose tissue of low (LRFI) and high (HRFI) residual feed intake pigs housed in good or poor hygiene conditions for six weeks.**

| | Good | | Poor | | P-values [a] | | |
|---|---|---|---|---|---|---|---|
| Genes | LRFI | HRFI | LRFI | HRFI | Hyg | Line | H × L |
| | | | Adipocyte differentiation | | | | |
| PPARG | 0.52 ± 0.07 | 0.37 ± 0.10 | 0.35 ± 0.07 | 0.46 ± 0.13 | 0.639 | 0.867 | 0.168 |
| | | | Lipid metabolism | | | | |
| FASN | 0.44 ± 0.08 | 0.34 ± 0.07 | 0.29 ± 0.04 | 0.16 ± 0.03 | **0.016** | *0.076* | 0.880 |
| SREBP1 | 0.59 ± 0.05 | 0.68 ± 0.08 | 0.57 ± 0.04 | 0.50 ± 0.08 | *0.097* | 0.893 | 0.203 |
| ME | 0.48 ± 0.08 | 0.36 ± 0.06 | 0.29 ± 0.03 | 0.20 ± 0.03 | **0.008** | *0.091* | 0.791 |
| | | | Lipid transport | | | | |
| LCN2 | 0.36 ± 0.06 | 0.22 ± 0.04 | 0.21 ± 0.03 | 0.13 ± 0.02 | **0.023** | **0.034** | 0.521 |
| | | | Carbohydrate metabolism | | | | |
| GLUT4 | 0.45 ± 0.07 | 0.37 ± 0.05 | 0.33 ± 0.05 | 0.27 ± 0.03 | *0.062* | 0.250 | 0.924 |
| | | | Mitochondrial metabolism | | | | |
| COX1 | 0.60 ± 0.03 | 0.65 ± 0.03 | 0.72 ± 0.05 | 0.62 ± 0.08 | 0.379 | 0.628 | 0.150 |
| | | | Oxidative stress | | | | |
| GPX3 | 0.25 ± 0.01 | 0.45 ± 0.04 | 0.23 ± 0.02 | 0.57 ± 0.09 | 0.246 | **<0.001** | 0.131 |
| | | | Adipokines | | | | |
| ADIPOQ | 0.60 ± 0.03 | 0.63 ± 0.04 | 0.59 ± 0.03 | 0.59 ± 0.07 | 0.544 | 0.765 | 0.702 |
| ADIPOR1 | 0.66 ± 0.03 | 0.61 ± 0.03 | 0.68 ± 0.04 | 0.63 ± 0.03 | 0.680 | 0.221 | 0.922 |
| ADIPOR2 | 0.72 ± 0.05 | 0.63 ± 0.05 | 0.62 ± 0.04 | 0.53 ± 0.05 | *0.068* | 0.110 | 0.993 |
| LEPTIN | 0.44 ± 0.09 | 0.37 ± 0.08 | 0.26 ± 0.11 | 0.17 ± 0.05 | *0.052* | 0.385 | 0.950 |
| IGF1 | 0.63 ± 0.04 | 0.64 ± 0.06 | 0.63 ± 0.03 | 0.64 ± 0.04 | 0.943 | 0.818 | 0.943 |
| TNF-α | 0.37 ± 0.04 | 0.53 ± 0.11 | 0.31 ± 0.05 | 0.41 ± 0.06 | 0.182 | **0.049** | 0.612 |
| IL-10 | 0.23 ± 0.02 | 0.36 ± 0.11 | 0.33 ± 0.06 | 0.51 ± 0.18 | 0.261 | 0.157 | 0.797 |
| | | | Pathogen recognition | | | | |
| TLR2 | 0.47 ± 0.06 | 0.61 ± 0.10 | 0.45 ± 0.05 | 0.62 ± 0.07 | 0.872 | **0.033** | 0.808 |
| TLR4 | 0.38 ± 0.04 | 0.44 ± 0.07 | 0.63 ± 0.08 | 0.61 ± 0.05 | **0.004** | 0.831 | 0.579 |

Notes.

Values are means ± SEM ($n = 7 - 10$ pigs/experimental group).

[a] Probability values for the effect of hygiene conditions (Hyg), genetic lines (Line), and the Hyg × Line (H × L) interaction. Boldface highlights significant differences ($P \leq 0.05$) and italicized character shows a trend ($0.05 < P < 0.10$).

In PRAT, the relative expressions of genes involved in lipogenesis were lower ($P = 0.016$ for *FASN* and $P = 0.008$ for *ME*), or tended to be lower ($P = 0.097$ for *SREBP1*) in pigs housed in poor compared with good hygiene conditions whatever the RFI line. The relative expression of the *LCN2* gene, a gene related to lipid transport, was also lower ($P = 0.023$) in pigs housed in poor condition compared with those housed in good condition. Concerning the *TLR4* gene, its expression was higher ($P = 0.004$) in pigs housed in poor than in pigs housed in good hygiene conditions. Next, we found that the relative expression of the *GLUT4* gene ($P = 0.062$), *LEPTIN* ($P = 0.052$) and *ADIPOR2* ($P = 0.068$) tended to be lower in pigs housed in poor compared with pigs housed in good hygiene conditions. Then, the relative expressions of four other genes were found to differ between RFI lines. The mRNA levels of *LCN2* was lower ($P = 0.034$) in HRFI than in LRFI pigs. Among the genes encoding enzymes protecting cells from oxidative stress, we identified that the mRNA levels

**Table 3    Relative expression of genes in subcutaneous adipose tissue of low (LRFI) and high (HRFI) residual feed intake pigs housed in good or poor hygiene conditions for six weeks.**

| Genes | Good | | Poor | | P-values [a] | | |
|---|---|---|---|---|---|---|---|
| | LRFI | HRFI | LRFI | HRFI | Hyg | Line | H × L |
| *Adipocyte differentiation* | | | | | | | |
| PPARG | 0.60 ± 0.07 | 0.57 ± 0.09 | 0.44 ± 0.08 | 0.38 ± 0.12 | **0.049** | 0.611 | 0.849 |
| *Lipid metabolism* | | | | | | | |
| FASN | 0.48 ± 0.08 | 0.40 ± 0.10 | 0.41 ± 0.05 | 0.23 ± 0.04 | 0.132 | *0.093* | 0.504 |
| SREBP1 | 0.58 ± 0.05 | 0.65 ± 0.10 | 0.66 ± 0.05 | 0.59 ± 0.05 | 0.889 | 0.981 | 0.276 |
| ME | 0.46 ± 0.07 | 0.35 ± 0.06 | 0.37 ± 0.04 | 0.29 ± 0.06 | 0.174 | 0.102 | 0.824 |
| *Lipid transport* | | | | | | | |
| LCN2 | 0.44 ± 0.06 | 0.31 ± 0.03 | 0.42 ± 0.08 | 0.24 ± 0.02 | 0.443 | **0.011** | 0.687 |
| *Carbohydrate metabolism* | | | | | | | |
| GLUT4 | 0.54 ± 0.05 | 0.51 ± 0.08 | 0.51 ± 0.04 | 0.54 ± 0.05 | 0.976 | 0.955 | 0.642 |
| *Mitochondrial metabolism* | | | | | | | |
| COX1 | 0.49 ± 0.04 | 0.51 ± 0.04 | 0.65 ± 0.05 | 0.46 ± 0.05 | 0.299 | *0.082* | **0.039** |
| *Oxidative stress* | | | | | | | |
| GPX3 | 0.32 ± 0.03 | 0.54 ± 0.09 | 0.33 ± 0.03 | 0.71 ± 0.10 | 0.149 | **<0.001** | 0.190 |
| *Adipokines* | | | | | | | |
| ADIPOQ | 0.56 ± 0.03 | 0.58 ± 0.04 | 0.61 ± 0.04 | 0.57 ± 0.08 | 0.655 | 0.771 | 0.525 |
| ADIPOR1 | 0.69 ± 0.03 | 0.65 ± 0.03 | 0.80 ± 0.04 | 0.68 ± 0.05 | *0.078* | **0.043** | 0.294 |
| ADIPOR2 | 0.73 ± 0.05 | 0.70 ± 0.05 | 0.73 ± 0.03 | 0.65 ± 0.06 | 0.623 | 0.265 | 0.573 |
| LEPTIN | 0.29 ± 0.05 | 0.37 ± 0.08 | 0.18 ± 0.04 | 0.46 ± 0.12 | 0.879 | **0.022** | 0.217 |
| IGF1 | 0.70 ± 0.05 | 0.76 ± 0.05 | 0.67 ± 0.02 | 0.82 ± 0.05 | 0.690 | **0.030** | 0.328 |
| TNF-α | 0.51 ± 0.08 | 0.66 ± 0.06 | 0.51 ± 0.06 | 0.64 ± 0.12 | 0.901 | *0.099* | 0.878 |
| IL-10 | 0.36 ± 0.04 | 0.71 ± 0.13 | 0.32 ± 0.04 | 0.44 ± 0.16 | 0.118 | **0.016** | 0.216 |
| *Pathogen recognition* | | | | | | | |
| TLR2 | 0.44 ± 0.06 | 0.53 ± 0.05 | 0.45 ± 0.04 | 0.47 ± 0.06 | 0.653 | 0.323 | 0.523 |
| TLR4 | 0.56 ± 0.05 | 0.58 ± 0.07 | 0.44 ± 0.07 | 0.41 ± 0.07 | **0.037** | 0.928 | 0.794 |

**Notes.**

Values are means ± SEM ($n = 5 - 10$ pigs/experimental group).

[a] Probability values for the effect of hygiene conditions (Hyg), genetic lines (Line), and the Hyg × Line (H × L) interaction. Boldface highlights significant differences ($P \leq 0.05$) and italicized character shows a trend ($0.05 < P < 0.10$).

of *GPX3* were higher ($P < 0.001$) in HRFI pigs than in LRFI pigs. Similarly, the relative expression of the *TNF-α* ($P = 0.049$) and the *TLR2* ($P = 0.033$) genes were higher in HRFI pigs than in LRFI pigs. Finally, the other genes studied were not significantly impacted by hygiene conditions nor by the genetic line in the PRAT (Table 2 and Table S1).

In SCAT (Table 3), the relative expression of only two genes were significantly affected by housing conditions. The mRNA levels of *PPARG,* a gene involved in adipocyte differentiation, decreased in pigs housed in poor compared with pigs housed in good hygiene condition ($P = 0.049$). Regarding the genes associated with the recognition of pathogens, it appeared that the relative expression of *TLR4* decreased ($P = 0.037$) in pigs from poor compared with pigs from good hygiene conditions whatever the RFI line. Next, we found that the relative expressions of the *LCN2* gene, involved in lipid transport, were lower ($P = 0.011$) in HRFI pigs than in LRFI pigs. Furthermore, the relative expression

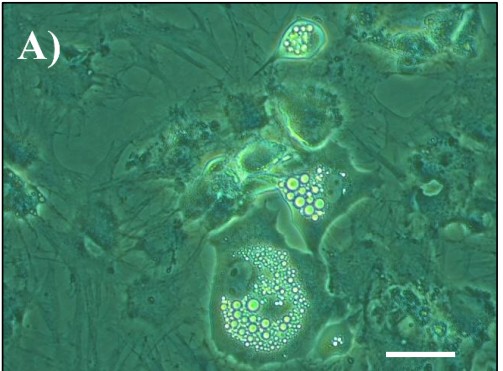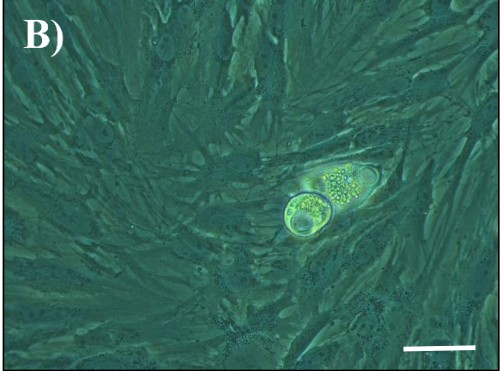

**Figure 1** **Photomicrographs of cultured cells at day 14 post seeding.** Cells from the stromal vascular fraction were isolated from subcutaneous adipose tissue of pigs at the end of the hygiene challenge and were cultured in adipogenic medium in the absence (A) or presence (B) of 10 ng/mL TNF-$\alpha$ ($\times$40, white bar stands for 10 μm).

of the *GPX3* gene was higher ($P < 0.001$) in HRFI pigs than in LRFI pigs. With respect to the relative gene expression of various adipokines and their receptors, we have shown that the mRNA levels of *LEPTIN* ($P = 0.022$), *IGF1* ($P = 0.030$) and *IL-10* ($P = 0.016$) were more expressed in HRFI pigs than in LRFI pigs whatever their housing conditions. In the same way, the relative expression of the *TNF-$\alpha$* gene tended to be increased ($P = 0.099$) in HRFI pigs compared with LRFI pigs. Next, the relative expressions of the *ADIPOR1* gene were lower ($P = 0.043$) in HRFI pigs than in LRFI pigs. Finally, the other genes studied were not significantly impacted by hygiene conditions nor by the genetic line in the SCAT (Table S2).

## Adipogenic differentiation of SVF cells isolated from SCAT in response to TNF-$\alpha$

Differentiation of SVF cells isolated from SCAT was determined on culture on D14, D18 and/or D22 post-seeding. At D14, D18 and D22, adipocyte differentiation of cells was not impacted by the genetic line, nor by the housing conditions of pigs ($P > 0.05$). Addition of TNF-$\alpha$ to the medium tended to decrease or decreased adipocyte differentiation at D14 ($P = 0.082$), at D18 ($P = 0.046$) and at D22 ($P = 0.009$) (Figs. 1 and 2).

## Whole blood cell response to LPS

In whole blood cells cultured in the absence or presence of LPS, the gene expression of *IL-1$\beta$*, *IL-6*, *IL-10*, *IL-15*, *TNF-$\alpha$*, *TGF$\beta$1*, *TLR2*, *TLR4*, *GPX3*, *ADIPOR1*, *ADIPOR2* and *LEPR* were not affected by hygiene of housing conditions (Table 4). The relative expression of the *GPX3* gene was higher ($P < 0.001$) in HRFI than in LRFI pigs. In addition, the relative expressions of the *IL-10* ($P = 0.027$), *TGF$\beta$1* ($P = 0.023$) and *ADIPOR2* ($P = 0.05$) genes were lower in HRFI than in LRFI pigs. The mRNA levels of *TGF$\beta$1*, *ADIPOR1*, *ADIPOR2* and *LEPR* were not influenced by the LPS stimulation in contrast to the other evaluated genes.

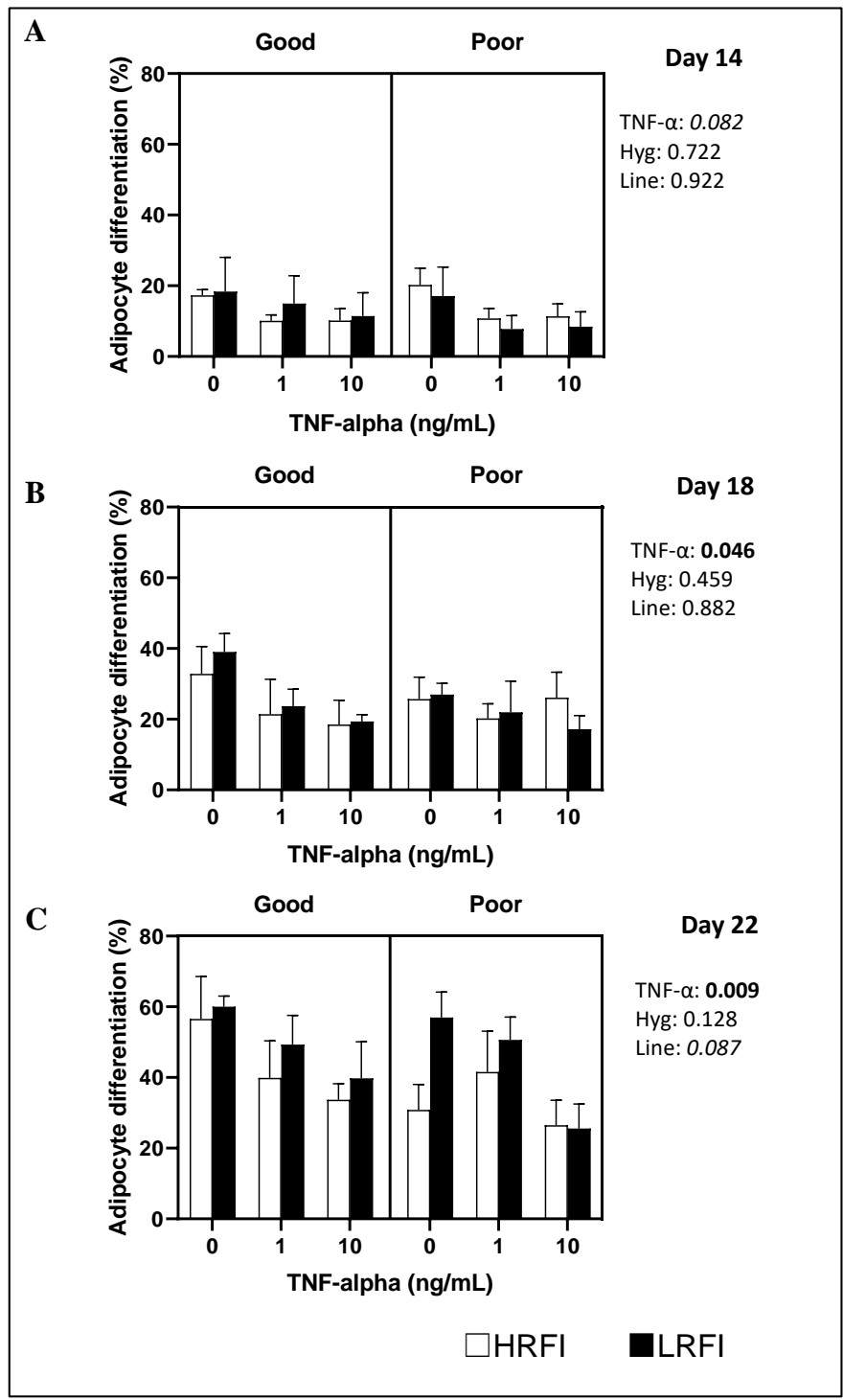

**Figure 2 Adipocyte differentiation of cells at Day 14, Day 18 and Day 22 post-seeding.** Cells from the stromal vascular fraction were isolated from subcutaneous adipose tissue of high (HRFI, white bar) and low (LRFI, black bar) residual feed intake pigs housed in good or poor hygiene conditions in the absence or presence of TNF-$\alpha$. Adipocyte differentiation are expressed as means $\pm$ SEM ($n = 3 - 5$ pigs/experimental group). Boldface highlights significant differences ($P \leq 0.05$) and italicized character shows a trend ($0.05 < P < 0.10$). All interactions are greater than 0.1 at Days 14, 18 and 22.

## DISCUSSION

In the current study, an experimental model of poor hygiene of housing conditions was used to induce a systemic inflammation thereby activating the immune system of LRFI and HRFI growing pigs, with LRFI pigs being more efficient than HRFI pigs, and modifying their metabolism (*Chatelet et al., 2018*; *Fraga et al., 2021*). Higher concentrations of plasma haptoglobin and of total IgG in animals placed in poor housing conditions confirmed that the health challenge induced a moderate inflammatory state in both lines of RFI pigs (*Chatelet et al., 2018*). Our major finding is that the responses measured in this study could explain, at least partially, the better ability of LRFI pigs to cope with such a challenge as previously reported (*Chatelet et al., 2018*). Indeed, the level of expression of the gene encoding the pro-inflammatory cytokine TNF-α was lower in the two adipose tissues studied (PRAT and SCAT) in LRFI pigs compared with HRFI pigs, whatever the hygiene conditions of housing. In addition, we observed that blood immune cells from LRFI pigs exhibited a higher baseline level of anti-inflammatory cytokine mRNA (*IL-10* and *TGFβ1*) compared with HRFI pigs. All of these data suggest that LRFI pigs secrete fewer pro-inflammatory and higher anti-inflammatory cytokines compared with HRFI pigs, which would help them to limit the deleterious effect of inflammation and therefore to maintain better their growth performance in response to this health challenge (*Chatelet et al., 2018*).

This study also indicates that the current challenge induced a depot-specific adipose tissue response with a greater number of genes affected by the hygiene challenge in PRAT than in SCAT. With the presence of Toll-like receptor 4 and 2, adipocytes and immune cells in adipose tissue are able to respond to pathogen structures such as bacterial LPS (*Lin et al., 2000*; *Asadzadeh Manjili, Yousefi-Ahmadipour & Kazemi Arababadi, 2020*). Thus, the finding of a greater expression of the *TLR4* gene in PRAT of pigs housed in poor hygiene conditions may be associated with an activation of TLR4 signaling pathways that are known to lead to the production of IL-6 and TNF-α, two proinflammatory cytokines. This latter hypothesis is consistent with a previous study showing that the TLR activation measured 6 h after a LPS challenge in young pigs was associated with an overexpression of pro-inflammatory genes in adipose tissue (*Guo et al., 2015*). Nevertheless, we failed to provide an increase in the expressions of these two proinflammatory cytokines in response to the current challenge. This suggests that PRAT either had a very limited inflammatory response to the hygiene challenge, or we missed a short inflammatory response that may have occurred earlier in PRAT. This latter hypothesis is supported by our previous findings (*Chatelet et al., 2018*) showing that the inflammatory response of animals was greater three weeks after the onset of the health challenge than after six weeks of exposure. In PRAT, we also observed a decrease in the expression of genes involved in fatty acid and cholesterol synthesis (*FASN, SREBP1* and *ME*) and lipid transport (*LCN2*). Similarly, *GLUT4* mRNA levels in PRAT tended to be lower in those pigs suggesting a decrease in lipogenesis in this adipose depot. These results are consistent with the finding of a lower mass of PRAT in pigs housed in poor conditions compared with pigs housed in good hygiene conditions. We hypothesized that pigs may allocate more energy to cope with the current challenge instead

**Table 4  Relative expression of genes in whole blood cells cultured in the absence (−) or presence (+) of LPS and obtained from low (LRFI) and high (HRFI) residual feed intake pigs housed in good or poor hygiene conditions for six weeks.**

| Genes | LPS | Good | | Poor | | LPS | Line |
|---|---|---|---|---|---|---|---|
| | | LRFI | HRFI | LRFI | HRFI | | |
| GPX3 | − | 0.76 ± 0.27 | 2.86 ± 1.28 | 0.46 ± 0.07 | 3.66 ± 1.32 | **0.011** | **<0.001** |
| | + | 0.34 ± 0.06 | 1.33 ± 0.38 | 0.26 ± 0.04 | 1.65 ± 0.42 | | |
| IL-10 | − | 0.29 ± 0.07 | 0.21 ± 0.05 | 0.35 ± 0.08 | 0.28 ± 0.07 | **<0.001** | **0.027** |
| | + | 3.23 ± 0.38 | 2.35 ± 0.49 | 3.10 ± 0.17 | 2.41 ± 0.32 | | |
| TGF β1 | − | 1.62 ± 0.26 | 1.15 ± 0.20 | 1.61 ± 0.20 | 1.37 ± 0.21 | 0.51 | **0.023** |
| | + | 1.64 ± 0.18 | 1.23 ± 0.26 | 1.79 ± 0.19 | 1.50 ± 0.16 | | |
| IL-6 | − | 0.93 ± 0.92 | 0.01 ± 0.005 | 0.04 ± 0.02 | 0.02 ± 0.005 | **0.024** | 0.357 |
| | + | 0.34 ± 0.10 | 1.12 ± 0.37 | 2.11 ± 0.83 | 0.89 ± 0.35 | | |
| TNF-α | − | 1.06 ± 0.42 | 0.60 ± 0.18 | 0.66 ± 0.05 | 0.61 ± 0.14 | **<0.001** | 0.753 |
| | + | 1.30 ± 0.18 | 2.87 ± 0.84 | 2.50 ± 0.97 | 1.93 ± 0.38 | | |
| IL-1 β | − | 1.08 ± 1.05 | 0.02 ± 0.006 | 0.04 ± 0.003 | 0.03 ± 0.003 | **<0.001** | 0.514 |
| | + | 1.30 ± 0.32 | 2.36 ± 0.84 | 2.59 ± 0.80 | 1.44 ± 0.46 | | |
| IL-15 | − | 3.29 ± 1.38 | 2.54 ± 0.64 | 3.54 ± 0.69 | 2.46 ± 0.28 | **0.001** | 0.400 |
| | + | 1.42 ± 0.24 | 1.57 ± 0.43 | 1.21 ± 0.22 | 1.32 ± 0.31 | | |
| TLR2 | − | 0.81 ± 0.24 | 0.53 ± 0.13 | 0.79 ± 0.15 | 0.75 ± 0.22 | **<0.001** | 0.254 |
| | + | 1.44 ± 0.28 | 1.13 ± 0.13 | 1.30 ± 0.20 | 1.24 ± 0.19 | | |
| TLR4 | − | 0.65 ± 0.18 | 0.41 ± 0.12 | 0.76 ± 0.18 | 0.53 ± 0.14 | **0.024** | 0.109 |
| | + | 1.00 ± 0.25 | 0.82 ± 0.15 | 1.03 ± 0.21 | 0.78 ± 0.16 | | |
| ADIPOR1 | − | 0.97 ± 0.25 | 1.08 ± 0.29 | 0.93 ± 0.23 | 1.38 ± 0.29 | 0.948 | 0.155 |
| | + | 1.14 ± 0.18 | 1.28 ± 0.34 | 0.85 ± 0.15 | 1.13 ± 0.20 | | |
| ADIPOR2 | − | 1.10 ± 0.23 | 0.84 ± 0.20 | 0.99 ± 0.17 | 0.80 ± 0.21 | 0.257 | **0.05** |
| | + | 0.92 ± 0.12 | 0.56 ± 0.07 | 0.93 ± 0.13 | 0.80 ± 0.15 | | |
| LEPR | − | 1.26 ± 0.45 | 1.07 ± 0.42 | 1.10 ± 0.24 | 1.56 ± 0.37 | 0.988 | 0.198 |
| | + | 0.55 ± 0.14 | 0.85 ± 0.22 | 0.86 ± 0.17 | 2.75 ± 1.64 | | |

**Notes.**

Values are means ± SEM ($n = 5 - 9$ pigs/experimental group).

[a]Probability values for the effect of the LPS treatment (LPS) and the genetic lines (Line). There was no significant effect of hygiene conditions and no significant interactions.

Boldface highlights significant differences ($P \leq 0.05$) and italicized character shows a trend ($0.05 < P < 0.10$).

of storing it as fat in the PRAT. These changes were also associated with greater activities of antioxidant enzymes in the PRAT of pigs housed in poor hygiene conditions which supports the occurrence of oxidative stress (*Sierzant et al., 2019*). A more extensive investigation of gene expressions (*e.g.*, RNA-seq) is required to further support the idea that PRAT is more responsive to the hygiene challenge than SCAT and to identify the main biological processes affected by the hygiene challenge. Furthermore, other tissues/organs than adipose tissue may respond to hygiene conditions. The study of skeletal muscle in these animals at the end of the hygiene challenge is consistent with the presence of an inflammatory state within this tissue (*Quéméner et al., 2022*). Indeed, pigs housed in poor hygiene conditions, regardless of the RFI line, had a higher proportion of CD45[+] hematopoietic cells than pigs housed in standard conditions. Two recent studies from the same group based on the investigation of transcriptomes of pigs differing in feed efficiency further support the

role of liver and muscle in the response to inflammatory stimuli (*Horodyska et al., 2018*; *Horodyska et al., 2019a*).

In SCAT, hygiene conditions decreased *TLR4* expression, and did not influence the expression of genes of lipogenesis, which is consistent with the lack of change in the mass of this tissue at the end of the challenge. In agreement with observations in PRAT, there was no influence of hygiene conditions on the mRNA levels of *TNF-α* in SCAT. As TNF-α is mainly produced by monocytes and macrophages, our recent finding of a similar proportion of hematopoietic cells, identified as CD45$^+$ cells, in SCAT of the four experimental groups (*Quéméner et al., 2022*) further support the idea that an inflammatory response cannot be detected in SCAT at the end of the challenge used in this study. Besides, the current findings further illustrate the differences in the expression of genes between SCAT and PRAT which is located within the abdominal cavity. In many species (*Tchkonia et al., 2013*) including pigs (*Gondret et al., 2016*), differences in adipogenic factors, responses to hormones, metabolic properties or secretion of inflammatory cytokines have been reported between SCAT and visceral adipose tissue. With respect to genes related to inflammation and immune response, several examples related to human obesity, demonstrate that visceral adipose tissue is more impacted than SCAT (*Harman-Boehm et al., 2007*; *Kranendonk et al., 2015*). In these studies, some differences related to adipocyte sizes, capillary densities and secreted amounts of adipokines have been reported. Therefore, during a moderate but long-lasting inflammatory challenge, PRAT would be mobilized to support the inflammatory response, while SCAT would be spared.

The observed decrease in the relative weight of PRAT in both RFI lines (*Sierzant et al., 2019*) was associated with a reduction in the level of expression of the *LEPTIN* gene in PRAT of pigs housed in poor hygiene conditions compared with those housed in good hygiene conditions. This hormone is primarily produced and secreted by adipocytes, and is involved in the regulation of feed intake and energy expenditure (*Barb, 1999*). The decrease in PRAT mass could contribute to the decrease in leptinemia leading to resumption of feed intake. Nevertheless, we have shown that the fasting plasma concentrations of leptin did not differ significantly between the four investigated groups (*Vincent et al., 2022*). Since SCAT is the main adipose depot in pigs, our results suggest that its contribution to leptinemia may be more significant than that of PRAT. Thus, leptinemia, being a marker of fatty tissue quantity in the body (*Kalina et al., 1999*), could be considered as a good marker of subcutaneous adiposity.

Further investigations were undertaken to determine whether the *in vitro* responses of adipose tissue and blood cells to pro-inflammatory molecules were impacted by the hygiene challenge in pigs of the two RFI lines. Our study clearly shows that TNF-*α* induced a reduction of adipogenic differentiation of FSV cells isolated from SCAT as shown in previous studies (*Petruschke & Hauner, 1993*; *Xu, Sethi & Hotamisligil, 1999*). It further indicates that the response to TNF-α was similar in cells from the four experimental groups despite the findings of a higher proportion of a population of mesenchymal stromal/stem cells (CD45$^-$CD56$^-$ cells) in SCAT from pigs housed in poor hygiene conditions compared with pigs housed in good hygiene conditions (*Quéméner et al., 2022*). Nevertheless, with the current available data, it is impossible to know whether there are differences in

differentiation ability between cell populations. Further studies need to be performed, such as sorting SVF cell populations before their culture *in vitro*. Similarly, there was no significant effect of hygiene conditions on the response of cultured blood cells to LPS. We cannot exclude the hypothesis that we missed a change in the properties of cells that may have occurred earlier.

Our study further shows that the expression of several genes in adipose tissues or in whole blood cells differed between HRFI pigs and LRFI pigs regardless of their housing conditions. We clearly found a higher expression of the gene encoding the GPX3 enzyme in PRAT, SCAT and in blood cells in HRFI pigs compared with LRFI pigs. This enzyme is involved in the detoxification of reactive oxygen species. This clear and systematic difference between the two RFI lines have been observed previously in muscle and blood (*Vincent et al., 2015*; *Jégou et al., 2016*). Because isolated adipocytes from HRFI pigs also produced more reactive oxygen species (ROS) (*Sierzant et al., 2019*), we hypothesize that HRFI pigs produced more antioxidant molecules in response to increased ROS production, potentially deleterious to cells, in order to maintain their redox balance (*Radak, Chung & Goto, 2005*). However, the expression level of the *GPX3* gene was not modified in response to the hygiene challenge in the two examined depots of adipose tissue. This contrasts with our previous work (*Sierzant et al., 2019*) showing a clear response of the anti-oxidant system of pigs, *e.g.*, a greater activity of several antioxidant enzymes (glutathione reductase, superoxide dismutase and catalase) in the PRAT of pigs housed in degraded housing conditions. Thus, the present result for *GPX3* indicates either that the response to the hygiene inflammatory challenge did not occur at the transcriptional level for this enzyme, or that GPX3 was only involved in the maintenance of the basal redox status but not in the response to a challenge.

The present study shows that there were also line differences on the expression of genes related to inflammation. Whole blood cells from HRFI pigs in culture expressed the same level of pro-inflammatory cytokines (*IL-1β*, *IL-6*, *TNF-α*) and toll-like receptors (*TLR2* and *4*) as cells from LRFI pigs, but had lower expression of *IL-10* and *TGF-β*, two cytokines involved in the down-regulation of the inflammatory response compared with cells from LRFI pigs (*Taylor et al., 2006*). With the current observations and a similar expression of *LEPTIN* and *ADIPOR1* genes in blood cells of pigs from both RFI lines, we cannot determine whether the adipose-to-immune tissue communication mediated these line effects. In PRAT, compared with LRFI pigs, HRFI pigs expressed higher mRNA levels of *TLR2*, whose activation triggers a pro-inflammatory signal, and of TNF-α, one of the main inflammatory cytokines mediating the metabolic effects of inflammation (*Gabler & Spurlock, 2008*; *Ajuwon, 2014*). Once again, effects were less clear-cut in SCAT, where the gene expression increased for *IL-10* and tended to increase for *TNF-α* in HRFI pigs compared with LRFI pigs. Altogether, these results suggest a greater pro-inflammatory state in adipose tissues of the HFRI line in comparison with the LRFI line. This difference between the two lines might play a role in the difference in feed efficiency, as inflammation impairs adipose tissue accretion (*Gabler & Spurlock, 2008*), and in robustness to environmental aggressions, as too much inflammation is counterproductive for resistance to microbial infections and diseases resolution (*Levy & Serhan, 2014*). This is what was observed with the present model of inflammation, since HRFI pigs had greater score of lung lesions and

a greater prevalence of pneumonia, whatever their hygiene environment (*Chatelet et al., 2018*).

## CONCLUSIONS

The current study indicates that the impact of hygiene of housing on the expression of genes in adipose tissues and on the features of SCAT adipose cells as well as whole blood cells in response to TNF-α or LPS was similar in LRFI and HRFI pigs despite differences in the expression of some genes like *GPX3* between the two lines. It further shows a depot-specific response of adipose tissue to the challenge with a decrease in the expression of genes related to fatty acid synthesis (*FASN* and *ME*) in PRAT but not in SCAT in response to the hygiene challenge. A larger investigation of gene expressions is required to confirm this depot-specific response. The complementary data obtained both *in vivo* and *in vitro* are not consistent with an inflammatory response in adipose tissues and blood cells at the end of the challenge. Further studies are needed to evaluate the hypothesis that adipose tissues and blood cells may have been affected earlier.

### Funding

The research leading to these results has received funding from the European Union's Seventh Framework Programme for Research, Technological Development and Demonstration (grant number 613574, PROHEALTH project) to support animal costs and from INRAE to support analytical measurements costs. Audrey Quéméner was supported by a PhD scholarship from INRAE (Phase division) and the research fund of Région Bretagne (France). Funders approved the general objectives of the study but have no roles in its design and data collection nor interfered with data interpretation and conclusions. The funders had no role in study design, data collection and analysis, decision to publish, or preparation of the manuscript.

### Grant Disclosures

The following grant information was disclosed by the authors:
European Union's Seventh Framework Programme for Research, Technological Development and Demonstration: 613574.
European Union's Seventh Framework Programme for Research, Technological Development and Demonstration: 613574.

### Competing Interests

The authors declare there are no competing interests.

### Author Contributions

- Audrey Quéméner performed the experiments, analyzed the data, prepared figures and/or tables, authored or reviewed drafts of the article, and approved the final draft.

- Marie-Hélène Perruchot conceived and designed the experiments, performed the experiments, authored or reviewed drafts of the article, and approved the final draft.
- Frédéric Dessauge performed the experiments, analyzed the data, prepared figures and/or tables, authored or reviewed drafts of the article, and approved the final draft.
- Annie Vincent performed the experiments, analyzed the data, authored or reviewed drafts of the article, and approved the final draft.
- Elodie Merlot conceived and designed the experiments, analyzed the data, authored or reviewed drafts of the article, and approved the final draft.
- Nathalie Le Floch conceived and designed the experiments, authored or reviewed drafts of the article, and approved the final draft.
- Isabelle Louveau conceived and designed the experiments, prepared figures and/or tables, authored or reviewed drafts of the article, and approved the final draft.

## Data Availability

The data are available at Louveau, Isabelle, 2022, "Gene expressions in porcine adipose tissues and blood cells", https://doi.org/10.57745/OTCFFO, Recherche Data Gouv, V1, UNF:6:wJxmAdLXcqnXx8VF0nG7vA== [fileUNF].

## Supplemental Information

Supplemental information for this article can be found online at http://dx.doi.org/10.7717/peerj.14405#supplemental-information.

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
