# Peer review of "Hygiene of housing conditions and proinflammatory signals alter gene expressions in porcine adipose tissues and blood cells"

_PeerJ, doi:10.7717/peerj.14405_

## Round 0.1 · original submission · Major Revisions

I agree with Reviewer 2 regarding the validity of the findings and misleading sentences. There are no details in the method section about how the genes were chosen to validate the claim and associate the immune pathways. This generalized claim can only be inferred from an unbiased screening like RNA sequencing or validating several targeted pathways. These aspects should be reasoned in the manuscript and conclusions along with clearly stating them as a limitation of the study.

Reviewer 1 ·

Basic reporting

1. Line 23. Can you provide a few lines of evidence showing that genetic selection causes the development and metabolism of adipose tissue? I would rather claim that “the metabolism of adipose tissue is associated with feed efficiency”.
2. It would help read the paper if the authors briefly introduced the background of all differently expressed genes before showing the results.
3. Line 224, “the pig was used as a random factor” What does it mean?
4. A few typos: line 216” Statistical analyzes” ; line 399 “either that”.

Experimental design

1. In my opinion the biggest problem of this paper is that only a few genes were examined by qRT-PCR. Although these genes represent most significant ones regulating inflammation and immune response, the complex nature and coordination between multiple pathways requires a more general and systematic evaluation of the expression profiles in adipose tissue (e.g. RNA-seq). Accordingly, I believe some claims are misleading such as “It further demonstrates that PRAT was more responsive to the hygiene challenge than SCAT” in the abstract and “a decrease in the expression of genes related to fatty acid synthesis” in the conclusion part. The authors may want to confine these claims to specific genes.
2. How the exposure duration of 6 weeks was determined? Will a longer exposure time affect the results?
3. Hygiene is quite a subjective concept. If used in a research paper as a variant, I suggest the authors define the good and poor housing conditions in great details so that others can better understand and repeat. Is it possible to identify typical environmental microbes in each condition?

Validity of the findings

1. It is not a common way to show P-values separately from the columns. In addition, I don’t think it is a good idea to keep the TNF-α dose as fixed effect. It is clear that, in Figure 2C, the dose of TNF-α exhibits inconsistent results between HRFI and LRFI pigs housed in poor condition. A single P-value may misinterpret these results.

·

Basic reporting

The manuscript is clearly presented with adequate presentations of methods and results. No concerns were identified.

Experimental design

The experimental design involved two genetic lines with different propensities for fat accumulation and two different storage sites. The use of two hygiene challenges contributed to inferences about gene responses of these tissues to inflammatory responses. The objectives are well defined. A short-coming of the design, acknowledged and discussed by the authors, was the time-course of the experiment. Different responses may have been detected if the samples were collected at earlier periods than the 6 weeks reported.

Validity of the findings

Methods are adequately described and the number of observations are sufficient to allow sound statistical inferences.
As mentioned in the comments on experimental design, the later time of tissue collections may have altered detection of tissue responses.
Likewise, the responses to hygiene conditions may be mediated by tissues other than adipose, ie., liver, spleen, GALT, and bone marrow. Some mention of mediation of immune challenges by other tissues should be included in the discussion. Given the focus of the current manuscript is adipose tissue, multiple tissues in the body respond to immune challenges. Likewise, the potential for direct effects vs. indirect effects of substrate supply (triglycerides) to adipose tissue induced by changes in hepatic metabolism should be discussed.

L 279. Would have been interesting to have the same responses reported in the PRAT tissue. Other than the time/expense required were there obstacles or reasons for not assaying cells isolated from PRAT? This is especially an interest in context of the inferences stated in L 310 to 311

Additional comments

4. Overall, the conclusions are confined to results reported and clearly presented. A very interesting manuscript with distinctions between subcutaneous and abdominal tissues .

---

## Round 0.2 · accepted · Accept

All concerns of the reviewers were adequately addressed and the amended manuscript is acceptable now.

Reviewer 1 ·

Basic reporting

no comment

Experimental design

no comment

Validity of the findings

no comment

·

Basic reporting

Acceptable changes have been made in response to the first review

Experimental design

see above

Validity of the findings

see above

Additional comments

Acceptable response were made by the authors